# Design and Performance Analysis of a Constellation of Nanosatellites to Monitor Water Quality in the Southern Catchment of the Baltic Sea

**DOI:** 10.3390/s23136192

**Published:** 2023-07-06

**Authors:** Karolina Kwapień, Patrycja Lasota, Michał Kędzierski, Piotr Walczykowski

**Affiliations:** Department of IMINT, Faculty of Civil Engineering and Geodesy, Military University of Technology, 00-908 Warsaw, Poland

**Keywords:** constellation of nanosatellites, orbit optimisation, remote water quality monitoring

## Abstract

The quality of inland waters has a significant influence on human life and the functioning of the environment. The disasters that result from water pollution may cause major financial losses and lead to irreversible changes in the ecosystem, such as the dying out of endemic species of plants and animals. Quick detection of pollution sources may minimise those negative effects and reduce the costs of their elimination. The study presents a constellation design that provides imagery in the optic range and that might supplement the point water quality measurements that are conducted in situ. The area of interest was the southern catchment of the Baltic Sea and the main rivers in the region. The requirements for the designed mission were defined in reference to the remote sensing needs concerning the monitoring of water quality, the characteristics of the analysed area, and weather conditions. Based on these requirements, the Simera Sense MultiScape100 CIS sensor and the M6P nanosatellite manufactured by NanoAvionics were selected. The authors proposed a process for selecting the optimum orbit, taking into account the area of interest, the possibilities of the satellite platform, and of the sensor’s optics. As a result of the analyses, four concepts of creating a constellation were presented. Each constellation consisted of four nanosatellites. The designs were then subjected to performance analysis, considering the lighting limitations. Among the proposed systems, the constellation designed by the authors was distinguished; it used four orbital planes and achieved the coverage and availability of imagery in the time that was best suited to monitoring the waters. Thanks to a small number of platforms, the costs of the mission are relatively low, and it might significantly improve awareness of the current state of surface waters in the southern catchment of the Baltic Sea.

## 1. Introduction

Seas and oceans account for 97% of the global water resources. The rest is fresh water stored mainly in glaciers and ice fields. Nevertheless, it is the inland waters (0.6% of global resources) that are a source of potable water, are used in the water supply systems and in industrial processes, and which are therefore one of the most important ecosystems for human life. Due to the role and influence of inland water bodies on the environment and the economy, their quality is constantly monitored. The main aim of monitoring is to provide information that is necessary for water management in catchments, and to protect them from anthropogenic threats and eutrophication. Traditional methods of assessing the quality of surface waters are based on collecting and analysing water samples, usually no more often than once per month [1,2]. Although this approach provides accurate knowledge about the ecological and chemical status of rivers and water reservoirs, it generates high costs, is time consuming, and does not provide continuous monitoring and information on a regional scale.

The suspicion that monthly measurements do not take into account the quick changes that affect water quality inspired NASA to start work on the SeaWiFS project [3]. During the project, in the years 1997–2010, an electro optical sensor mounted on the OrbView-2 satellite was used to assess the quality of ocean and coastal waters. Another proof of the usefulness of satellite Earth observation techniques in water management is the project of the European Space Agency (ESA) that has been conducted since 2012. In this project, information obtained from the orbit are then used in managing water resources [4,5]. The SatBałtyk system was created in order to address the most complex problems of diagnosing the dynamic environment of the Baltic Sea [6,7]. The system uses measurement data obtained from over twenty sensors mounted on more than ten satellites [8].

However, these projects use classic satellite platforms that are usually equipped with sensors of a relatively low spatial resolution and focus mainly on monitoring seas and oceans. The manufacturing costs of a classic platform may be several times higher than the value of a single nanosatellite [9], and the temporal resolution of systems is usually approx. 1 day. Because of this, it is worth considering the application of smaller satellite platforms consisting of a larger number of satellites to create a constellation that would enable the monitoring of surface waters with a higher frequency thanks to its higher temporal resolution and adequate spatial resolution.

In the literature, the idea of using a constellation of nanosatellites to monitor the waters was discussed in the context of the design of an integrated satellite–terrestrial system based on the classical Walker’s concept that would generate lower costs than the traditional methods of monitoring the pollution status [10]. Similarly, the Walker constellation was used—taking into account the analysis of the influence of various orbit parameters on the temporal resolution—to monitor illegal immigration in the basin of the Mediterranean Sea, where the achieved revisit period was lower than 25 min [11]. Publications from the last 10 years return to the topic of using small and relatively cheap CubeSat satellites to monitor natural and anthropogenic phenomena that affect human life. One of the examples is the design and optimisation of the asymmetric constellation of CubeSat based on orbits with repeating ground tracks, which ensures continuous coverage of a specific area of interest to monitor hurricanes [12]. For the purposes of managing natural disasters in Brazil, the methodology of designing a CubeSat constellation that uses five orbital planes of different values of the ascending node was employed [13]. Another project that was initiated to address the problems with water management in Africa is the SWEET CubeSat system equipped with a hyperspectral sensor of a spatial resolution of 250 m that provides data about the state of the pollution of large freshwater lakes [14]. However, the ground pixel level makes it impossible to observe smaller inland water bodies. Effective monitoring of water bodies such as rivers would have to be based on data with a resolution of several meters. This is confirmed by the effective use of the 3-m images obtained by the existing PlanetScope constellation to determine the rate of flow in selected Siberian and Alaskan rivers [15].

The subject of this article is the design and performance analysis of a constellation of nanosatellites to monitor water quality in the southern catchment of the Baltic Sea The main aim is to create a concept of a satellite imaging mission that would meet the requirements of water control at a relatively low cost. The comprehensive approach refers not only to the studies on the selection of the optimum orbit and constructing the constellation, but also focuses on the imaging issues. The authors assumed that a precise definition of the needs that the mission will have to meet will allow for the selection of a specific satellite platform and sensor, and conducting analyses considering the specificity of these elements will provide insight into the actual possibilities of such a constellation.

## 2. Remote Sensing Monitoring of Inland Waters

Broadband remote sensing satellites have been used in the analyses of open sea and ocean waters since the 1970s, when the first satellites from the Landsat series were launched to the orbit. However, inland waters are a more demanding area of study, both in terms of spatial resolution and of the recorded spectral ranges. The difficulties are caused not only by the relatively smaller dimensions of freshwater bodies, but first of all by the high biological and optical complexity, the presence of suspensions, and the high dynamics of the occurring changes [16,17]. The breakthrough that enabled broader use satellite data in analysing inland waters was the emergence of more advanced super- and hyperspectral sensors (ASTER, ENVISAT, CHRIS) [17].

Remote sensing technologies enable the monitoring of water quality status in terms of the pollutants that change the optical properties of water. The main quality parameters of water that may be monitored remotely include transparency, turbidity, and the content of chlorophyll and organic matter, including carbon [8].

The light transmission capacity of water is determined with use of the transparency parameter, which is strongly linked to the presence of colloidal particles and suspensions. This is the main parameter used in assessing the quality of water. The most common method of measuring water transparency in remote sensing is the so-called Secchi disc method. The analysis of publications on the studies of the usability of this method revealed that the most commonly used wavelengths are approx.: 490 nm, 520 nm, 670 nm, 706 nm, and 750 nm [17]. However, formulae were also developed to successfully determine the transparency of water based on broad channels in the range of blue range (0.40–0.50 µm) and green range (0.50–0.60 µm), blue range (0.40–0.50 µm) and red range (0.60–0.70 µm), or green range (0.50–0.60 µm) and red range (0.60–0.70 µm) [16].

Another parameter is water turbidity, which determines the degree of light scattering and absorption by the particles contained in water. High water turbidity in itself does not pose a direct threat to human health, but it might signal a failure of the water treatment infrastructure or the presence of pollutants. To determine this parameter with the use of remote sensing methods, spectral indicators are used that operate most commonly on waves in the blue, red, and near infrared ranges, as these ranges are the most strongly scattered on suspension particles. An example is the mapping of the turbidity of the Klang River in the Malay Peninsula with use of just one spectral channel of the SPOT 5 satellite in the near infrared range (0.78–0.89 μm) [18].

Another very important index in surface water management is the content of chlorophyll-a. High concentration indicates the presence of cyanobacteria and is directly correlated to the adverse phenomenon of eutrophication. The periodical presence of cyanobacteria in the coastal waters of the Baltic Sea and in inland water bodies in Central and Eastern Europe is a serious problem [19]. The content of chlorophyll-a on the level of 8–10 mg/m^3^ in closed freshwater bodies indicates the process of cyanobacteria bloom, and when the level of 30 mg/m^3^ is exceeded, the water is unfit for drinking [20]. The spectral characteristics of chlorophyll-a shows a high absorption of radiation in the 450–475 nm and 670 nm ranges and high reflectance for the ranges of approx. 550 nm and 700 nm [16]. As far as lakes and rivers are concerned, a problematic issue is the presence of suspensions originating from land that are not correlated optically to chlorophyll, which hinders the remote sensing of its concentration in these waters. Studies on the applicability of specific spectral ranges in determining the presence of chlorophyll in inland waters have been conducted since the late 1990s and revealed a particular sensitivity of the bands near 713 nm and 667 nm [21]. Spectrometric measurements conducted from orbital altitudes with the use of the MERIS instrument were the basis for the successful development of various algorithms for the detection of chlorophyll-a in turbid waters based on narrow-band data [22]. On the other hand, recent studies on the usability of multi-channel data of low spectral resolution with the use of the WASI inversion methods in the remote sensing assessment of the concentration of chlorophyll have brought positive results. However, this method requires using predefined parameters for various types of waters [23].

High chlorophyll content indicates the presence of such phenomena as cyanobacteria bloom, the development of algae, and the overgrowing of water reservoirs. However, these processes may be predicted and counteracted early if there is an increased content of dissolved organic matter (DOM), whose main component is dissolved organic carbon (DOC). Satellite hyperspectral data have a great potential of assessing the level of DOM, which was confirmed by studies presented in recent publications, based on such sensors as Hyperion [24], MODIS [25], and MERIS [26]. Promising correlations between the level of DOM and the combinations of multispectral channels of the Landsat 8 satellite were also noticed in the preliminary studies on the usability of broadband data. These studies revealed the special usability of the channel in the 433 nm–453 nm range [27].

The above review confirms that analysing the phenomena that occur in water resources that have a significant influence on human life and the functioning of the environment requires obtaining information in the adequate ranges of the electromagnetic spectrum. Bands in the visible and near infrared range are particularly useful. The referenced examples allow us to state that in spite of the high optical complexity of inland waters, several correctly selected spectral channels should enable us to conduct monitoring of the main quality parameters of water.

## 3. The Objectives and Requirements of the Satellite Mission

The main aim of the mission is to provide data for the monitoring of inland water quality in the southern catchment of the Baltic Sea while maintaining low costs of the project. The research environment and the objectives and requirements for the designed mission, in reference to the planned goal, are described in the following sub-sections.

### 3.1. Area of Interest

The Baltic Sea catchment covers an area of more than 1,700,000 km^2^, and it is inhabited by approx. 85 million people. Considering the population density (Figure 1), the intensity of agriculture, and the level of industrialisation, the waters in the southern part of the catchment are the most exposed to anthropogenic pollutants and their poor quality affects a larger number of recipients.

Due to the above, the selected area of interest was the southern and southeastern part of the Baltic Sea catchment. Special attention was paid to the main rivers, which include two rivers of a longitudinal course (Vistula and Odra Rivers) and two latitudinal rivers (Niemen and Daugava). The areas adjacent to their estuaries belong to the most polluted parts of the Baltic Sea [28]. The linear nature of watercourses poses a challenge for satellite monitoring, which is better adapted to providing images of point targets. For the purposes of further analyses, the focus area was defined for the planned satellite mission. The area is contained between the following coordinates: 13° E–33° E and 49° N–58° N, and the main rivers with 2 km wide buffer zones whose axes are consistent with the streamline (Figure 2).

### 3.2. The Requirements of the Satellite Mission

In order to maintain the low costs of the project, it was assumed that nanosatellites placed on a Low Earth Orbit (LEO) would be used. To ensure constant altitude of the orbit and the resulting advantages for images, including constant spatial resolution, orbital speed, and the same orbital period of the satellites in the constellation, a circular orbit was selected.

Then, the issue of spatial resolution was considered. As a result of the image discretization process, a single pixel carries information about the averaged value of the intensity of electromagnetic radiation from an area equal to its ground size. If the dimensions of the object are smaller than GSD (Ground Sampling Distance), the image shows the averaged spectral response of the object and its surroundings. There are known cases when, in situations of high contrast with the background, it is possible to detect sub-pixel targets (smaller than the ground size of the pixel), yet the aim of this study is not to detect the surface waters but to ensure the monitoring of their status. The problem will remain unsolved if the size of the pixel is similar to the size of the water bodies, particularly in the case of rivers due to their linear nature (Figure 3).

Considering the presented state of affairs, it was determined that in order to eliminate the influence of coastal objects on the obtained water quality result to the greatest possible extent, the value of GSD should equal at least three times the width of the river. To this end, the average width of the beds of the main rivers of the Baltic Sea catchment was determined for specific sections (Table 1).

The width of watercourses in their most upstream parts starts from approx. 10 m. The sections closest to the source were omitted, as it is assumed that the likelihood of the presence of pollutants is much lower there. Considering the obtained values and the review of the possibilities of optical electronic sensors dedicated to nano class satellite platforms, the optimum required GSD value was considered to be 5 m. This size should ensure the appropriate representation of the downstream, midstream, and, in most cases, the upstream sections of the analysed rivers.

In reference to the review presented in Section 2, the sensor should be multispectral and provide imaging not only in the visible range but also in the near infrared range. The minimum requirements for the number of channels were defined as at least three channels in the VIS range and two in the NIR range.

The temporal resolution of the system should ensure knowledge about the quality status of the waters on an ongoing basis. It is assumed that in order to meet this objective, the period between subsequent revisits over the objects of interest should be from 1 to 7 days (Figure 4). As far as passive sensors are concerned, the temporal resolution results not only from the capacity of the constellation. It may also be decreased by lighting and cloud cover. For the planned mission, a 3-day revisit period in the spring and summer seasons was assumed, taking into account the lighting conditions, so that the potential cloud cover would not cause interruptions in data collection that last longer than one week (Table 2).

## 4. Methodology of Designing the Satellite Imaging Mission

The authors propose a methodology for determining the optimum orbit and designing a constellation of satellites for monitoring a specific area of interest based on the assumptions that low costs are maintained and that the required spatial–temporal resolution is achieved. The sequence of the stages of designing the constellation is presented in the diagram in Figure 5.

First of all, the geographical position of the area of interest must be defined. The required spatial and spectral resolution, as well as the revisit time, must be determined in reference to the objectives of the mission. The methodology assumes that it is a low-cost mission, which is already considered at the stage of selecting the imaging sensor. The proposed method of determining the altitude of a low circular orbit assumes that the optics of the selected sensor will be taken into account and that the required GSD will be achieved for the maximum possible inclination of the platform from the nadir. Then, the inclination is selected to ensure the largest possible coverage of the area of interest by the satellite. Defining the maximum altitude and selecting the optimum inclination allow us to reduce the number of satellites at the design stage, which enables us to lower the mission costs.

### 4.1. Altitude and Inclination

Higher orbit altitude means a shorter revolution period of the satellite and a larger field of regard for the sensor, but it also leads to lowering the spatial resolution. Because of this, it was assumed that the optimum solution would involve using the highest possible orbit altitude h, which, considering the specificity of the selected sensor and the platform inclination capacity, would ensure thar the required GSD is obtained (Figure 6).

The total field of regard (FOR) in which the satellite is able to target the field of view (FOV) is defined by the angle $\theta_{R}$ being the sum of the following:(1)$$\theta_{R}=\theta_{FOV}+2\theta_{off-nadir}.$$

The highest GSD will occur at the maximum possible deflection of the optical axis from nadir. The adopted ${GSD}_{max}$ value should be the minimum resolution required for the mission, which will occur for angle α (2). The value of the angle α consists of $\theta_{FOV}$ and $\theta_{off-nadir}$ resulting from the sensor and satellite platform used (3). Using trigonometric equations, reducing ${GSD}_{max}$ do ${GSD}_{nadir}$ allows us to calculate the altitude of orbit *h* after transformation (5).
(2)$${GSD}_{max}=\frac{{GSD}_{nadir}}{\cos\alpha }$$
(3)$$\alpha =\frac{\theta_{FOV}}{2}+\theta_{off-nadir}$$
(4)$${GSD}_{nadir}=h*\frac{px}{f}$$
(5)$$h=\frac{{GSD}_{max}*\cos\alpha *f}{px}$$

The determined orbit altitude enables us to calculate the values that are important for the imaging possibilities of the given sensor: the resolution obtained in nadir (4), the width of the imaging belt in nadir (6), and at maximum deflection from nadir (7).
(6)$${Swath}_{nadir}=2\tan\left(\frac{FOV}{2}\right)*h$$
(7)$${Swath}_{off-nadir}=h*\left[\tan\left(\alpha \right)-\tan\left(\theta_{off-nadir}-\frac{FOV}{2}\right)\right]$$

The next step was the determination of the optimum inclination of the orbit to the equator plane. Finding an inclination that ensures the largest coverage of the area of interest will enable us to achieve the required revisit time with a smaller number of satellites and thus to reduce mission costs. The authors considered using a polar heliosynchronous orbit for the determined altitude or an orbit with the inclination referring to the borders of the area of interest so that the satellite would make the return above the largest latitude that is important for the mission. The comparative analysis of the coverage achieved for the given inclinations may be performed based on the function of the density of probability distribution (8) of the coverage of random latitude x, treating the anomaly as a random variable from 0 to 2π [29].
(8)$$f\left(x\right)=\frac{\cos x}{\pi \sqrt{\sin^{2}i-\sin^{2}x}}\;\;dla(-i<x<i)$$

The eccentricity of the calculated orbit results from the assumption that a circular orbit will be used in the mission. The initial values of the rectascence and true anomaly of the satellite that vary in time are defined at the stage of designing the constellation.

### 4.2. Constellation Architecture

The aim of the final stage of designing the mission is to determine a structure of the constellation that will ensure the best conditions for observation of the targets of interest with the minimum number of satellites. For circular orbits of the same inclination, Walker’s constellation is commonly used in the literature. This concept, proposed by John Walker [30,31] ensures a symmetric structure thanks to the even distribution of all satellites and orbital planes. The constellation is defined by characteristic parameters: i: T/P/F, for which: i—inclination; T—total number of satellites in the constellation; P—number of orbital planes; and F—relative shift between satellites on neighbouring planes. The above parameters are used to calculate the angles: α,β,γ, which determine the relations between satellites that are moving on the same orbit and between neighbouring orbital planes (Figure 7).

Two configurations of Walker’s constellation may be distinguished: Delta, where the angles are determined from Equations (9)–(11); and Star, where 360° is replaced by 180°.
(9)$$\alpha =\frac{360{}^{\circ}}{P}$$
(10)$$\beta =360{}^{\circ}\frac{P}{T}$$
(11)$$\gamma =360{}^{\circ}\frac{F}{T}$$

The presented concept works well in satellite missions that require an even and relatively fast coverage of the areas of interest. The efficiency of specific architectures also depends on the nature and location of the targets. Therefore, it is worth considering modifying the concept and applying symmetric constellations designed by the authors in order to define the optimum arrangement of satellites for the specific mission.

## 5. Design of the Constellation of Nanosatellites

In the practical part of the research presented here, the design was subjected to efficiency analysis of the constellation of nanosatellites designated for the remote monitoring of the quality of surface waters in the southern catchment of the Baltic Sea. The project was realised in compliance with the presented methodology and in reference to the objectives and requirements of the mission described in Section 3.

### 5.1. Constellation Architecture

In reference to the objectives set for the satellite mission and the low-cost criterion, the electro-optic sensors dedicated to nanosatellites that are available on the market were reviewed. The authors considered using one of the following devices: Caiman Imager, MultiScape100 CIS, or HyperScape100. The technical specifications of the sensors are presented in Table 3.

The authors decided that the sensor that offered the optimum capacity and costs was the MultiScape100 CIS push broom scanner by Simera. The Caiman Imager camera has similar spectral resolution possibilities, although it additionally acquires the panchromatic channel. However, in water quality monitoring, this does not result in improved accuracy of pollutant detection. The GSD = 6 m at 500 km altitude ensured by the Caiman Imager would require lowering the orbit in order to achieve the spatial resolution required for the mission. As a consequence, the lifespan of the platform would be shortened. Other undesirable consequences include reducing the belt of imaging and the ground reach of the satellite, which would require using a larger number of platforms to achieve the required revisit time. Ultimately, this would increase the costs of the mission. Another considered device was the HyperScape100 hyperspectral sensor of a specification similar to that of MultiScape100 CIS. Considering the possibility to conduct efficient pollution monitoring based on several correctly selected hyperspectral channels (Section 1), it was decided that the increase in price resulting from choosing a more advanced hyperspectral sensor would be disproportionate to the improvement in capacity. Another aspect that supports the selection of MultiScape100 CIS was the fact that this sensor was used in the Napa-2 operational mission and that it was successfully tested in peri-terrestrial conditions [35].

The M6P satellite platform by NanoAvionics was chosen to carry theMultiScape100 CIS sensor [36]. The satellite meets the size requirement of 6 U, and it may carry loads of the dimensions up to 5 U. The platform is equipped with a triaxial stabilisation system using reaction wheels, so it has the possibility to manoeuvre and target the sensor with a speed up to 5°/s. The estimated nominal lifespan of the satellite and thus the duration of the designed mission is 3 years.

Currently, the most commonly used communication scheme for observational satellites is the use of the S and X bands. The S band is intended for two-way communication with the satellite platform in the field of control and tracking (so-called TT and C—Telemetry, Tracking and Command). This includes a ground-to-satellite link (uplink) for telecommands (TC) and a satellite-to-ground link (downlink) for receiving telemetry (TM). The X-band is used to send observation data from the satellite to the ground, so broadband transmission is used here, allowing for the transmission of large amounts of data.

The TT and C satellite link must be available regardless of the conditions and orientation of the satellite in the orbit. Therefore, narrowband transmission is used here in order to ensure the best possible link budget, which can be established even by smaller ground stations. Typical uplink speeds are in the range of 32–256 kbps and downlink speeds are 1–2 Mbps.

Observational data, on the other hand, needs a much wider bandwidth and higher transmission speeds. Due to the large amounts of data generated by modern observation instruments, it is necessary to use more effective transmission methods. DVB-S2 transmission is often chosen as a favourable method, allowing data transmission at speeds of up to several hundreds of Mbps, compared to direct QPSK/OQPSK modulation, where speeds of up to 50 Mbps are usually obtained. Even higher speeds are possible with the use of dual polarisation transmission. The DVB-S2 transmission also enables the use of the VCM (Variable Coding and Modulation) and ACM (Adaptive Coding and Modulation) modes, which allow for the adjustment of the transmission parameters to the current communication conditions. This allows for the more effective use of the entire communication window.

Due to the common utilization of LEO orbit by the most of observational satellites, the time of communication with the satellite for one earth station is limited to several communication windows per day (typically ~five windows of 10 min each) [37]. The number and time of communication passes also depends on the location of the station and the exact parameters of the orbit. This limits the commanding and transmission, especially for instruments that generate large amounts of data. To increase the access time, it is possible to use ground station network services (GSaaS—Ground Station as a Service), which have stations in various parts of the globe [38]. One of the best-known networks is KSAT (Kongsberg Satellite Services), which, in addition to having stations all over the world, owns and operates two uniquely located stations—one in Svalbard (SvalSat) and the other in Antarctica (TrollSat). Thanks to such a polar location, it is possible to maximise the service of SSO (Sun Synchronous Orbit) orbits, which are characterised by a high inclination; thus, the revisit over the polar regions is undertaken via each orbit revolution. Other examples of such networks are LeafSpace or RBC [39,40].

### 5.2. Orbit Parameters

Based on the method of calculating the orbit presented in Section 4.1., the maximum altitude that ensured obtaining 5 m GSD at the maximum deflection of the MultiScape100 CIS sensor was determined. Based on the manoeuvring system of the platform, the value of $\theta_{off-nadir}$ was assumed to be 20°. As a result of the applied calculation process, the orbit altitude was 491.88 km. Other important values were also calculated (Table 4).

During nadir imaging, the mission will enable us to acquire imagery with a resolution of 4.66 m.

Then, the influence of orbit inclination on the achieved coverage of the southern catchment of the Baltic Sea was analysed. According to the assumptions provided in the methodology, the analysed inclinations were:-*i* = 97.4°;-*i* = 58.0°.

The density of the probability distribution of coverage of random latitude per degree is presented in Figure 8. It is assumed that for the inclination of 58.0°, the ground track of the satellite crosses latitudes that are slightly smaller than the orbit inclination with the highest probability. Additionally, in comparison to the heliosynchronous orbit, the probability density for the area of interest (49°–58°) is much higher—approx. 2 times—and tends towards infinity for the upper limit.

The optimum value of the inclination was also calculated using simulations in the System Tool Kit (STK) software. Coverage was determined for the assumed FOR and diagrams were generated that showed the percentage of time spent on covering specific latitudes proportionately to the whole analysed working time of the satellite. The satellite placed on the orbit with the inclination of 58.0° achieved a higher time percentage of covering the latitudes of interest than the heliosynchronous satellite (Table 5). For the latitude of 49° these values were, respectively: 0.04% for i = 58.0°; 0.02% for i = 97.4°; for the latitude of 55°—0.07% for i = 58.0°; 0.03% for i = 97.4°.

The above analysis demonstrated that the inclination that ensured better coverage of the area of the southern catchment of the Baltic Sea was 58.0° (Figure 9 and Figure 10). The fact that the probability of coverage of the latitudes in the area of interest was approximately 2 times higher and the higher percentage of time spent above the southern catchment of the Baltic Sea make the 58.0° inclination more cost-effective. As a result, this value was adopted for the designed mission. The initial orbital parameters of the satellite that were the basis for building the constellation were as follows: a = 491.88 km; e = 0.00; i = 58.0°; ω = 0°; Ω = 0°; TA = 0°.

### 5.3. Coverage Analyses for a Single Nanosatellite

The usability of the selected orbit was assessed based on the analyses of coverage of the objects of interest by a single satellite in the System Tool Kit software. The analysis was conducted for July 2023, taking into account the lighting conditions. The simulation excluded time intervals that started one hour before dusk and those that ended one hour after dawn for each of the rivers in order to demonstrate the actual imaging capacity of the Simera Sense MultiScape100 CIS passive sensor. The results were presented in the diagram of imaging accessibility in time (Figure 11) and in graphic form showing the ground tracks of the satellite (Figure 12 and Figure 13).

The designed orbit enables us to acquire images of the main rivers in the southern catchment of the Baltic Sea with the use of one satellite, even up to three times per day, which is confirmed by the results of the first half of the analysis. However, due to the fact that the orbit is not heliosynchronous, the moments of imaging access shift in time, similarly to the position of the orbit in reference to the targets. As a result, in the second half of July 2023, some of the flights of the satellite fall in the period without sunlight so the temporal resolution falls below the required limit. This is an argument for designing a constellation that would be based on a selected orbit, which would enable us to use its advantages and to eliminate periods with no access to imaging.

### 5.4. Symmetric Constellation: Project 1

Based on the analysis of coverage for a single satellite and the criterion of maintaining low mission costs, the authors considered a set consisting of two platforms. Walker’s concept was used in the design, and it was assumed that the first satellite would be placed on the orbit discussed in the previous sub-section. The second satellite was placed on an orbital plane of the identical shape and size but displaced in rectascence (Ω) by 180° and, in the initial position, denoted by the true anomaly (TA) of 180°. As a result, the Project 1 symmetric constellation of the parameters 58°: 2/2/1 was created (Figure 14).

The designed constellation was assessed based on the access to river imaging. The shift of the second satellite in RAAN closed the gap in imaging that occurred in the second half of the analysed period, which is presented in Figure 15. The system permitted access to each of the rivers from two to four times per day.

The analysis was complemented by verifying the accessibility of imaging any section of each of the rivers with use of FOR in 3-day periods. The simulation took lighting into account by defining the minimum angle of elevation of the Sun of 10° for each point of the river. These results are presented in Table 6 and in Figure 16. During periods of access by only one satellite (1–3 July 2023, 29–31 July 2023), the constellation does not provide 100% coverage, particularly for the Vistula River, for which access was achieved, respectively, to 44.86% and 9.03% of its length. In the second analysed period, only 21.94% coverage was achieved for Odra and 74.08% for Daugava. During the effective flights of two satellites (15–17 July 2023), the Niemen River was not fully covered; access to 87.64% of its area was achieved. As a result, it is worth considering designing a larger constellation.

### 5.5. Symmetric Constellations Project 2–4

Considering the results obtained for Project 1, the authors decided to increase the number of satellite platforms to four. Four variants of constellation architecture were analysed. The constellations differed in terms of inclination (i), RAAN (Ω), and true anomaly (TA). In Project 2 and Project 3, the delta type of Walker’s constellation was used, with the following parameters: Project 2—58°: 4/4/0; Project 3—58°:4/2/1. The next two projects were based on the authors’ own concepts based on the orbit of i = 58°and the reverse orbit i = 122°. The position of orbital planes and nanosatellites in each constellation is presented in Table 7.

The constellations were assessed with use of the simulations, applying the same lighting limitations as those used in Project 1. The first analysis comparing the constellations concerned the access to imaging each section of the rivers. In the 3-day period, each project achieved 100% coverage with use of FOR, so the essential requirement of the mission was fulfilled.

Then, the number of accesses in time was analysed. The results are presented in Table 8. The differences in the access numbers were not sufficiently significant to reveal the most effective constellation, although the lowest values were usually obtained for Project 3.

Further on, the time required to achieve 95% coverage of the rivers with the use of FOR was determined. The results are presented in Table 9. In this case, larger differences between the projects were noted and the times achieved for specific rivers were also more varied. For Projects 2, 3, and 5, the periods of coverage for the Vistula River were relatively long. For Projects 3 and 5, 95% coverage was obtained only on the second days of the tests (respectively, after 36 h 19 min and 30 h 59 min), and for Project 2, only on the third day, after 54 h 01 min. The best results were obtained for the Project 4 constellation that obtained the required coverage for all rivers as early as in the first day.

The final analysis, presented in diagrams in Figure 17, Figure 18, Figure 19 and Figure 20 shows the imaging access for constellations and the percentage of coverage of all rivers that was achieved at the given moment. It is noted that for Project 4, the access was evenly distributed throughout the day, similarly to in Projects 2 and 5. Due to the influence of weather conditions, such a situation is more desirable than the accumulated access that was observed in Project 3.

To conclude the conducted research, it was found that the most appropriate constellation for the mission of monitoring the waters and main rivers in the area of the southern catchment of the Baltic Sea was Project 4, as proposed by the authors. In this case, the deciding factor was the evenly distributed access to all rivers. However, it should be noted that all variants met the main requirements of the mission (Figure 21).

## 6. Conclusions

The study presents the process of the design and analysis of a constellation of imaging nanosatellites designated for a specific mission which consisted in ensuring the monitoring of surface water quality in the southern catchment of the Baltic Sea. The main objective was to maintain low costs, so small satellite platforms were chosen. Considering the remote sensing needs related to the monitoring of inland waters, the authors proposed to use the Simera Sense MultiScape100 CIS multispectral sensor and the MP6 Nano Avionics platform. The study presents the two-stage process of designing the orbit that consisted of determining the altitude based on mission requirements and sensor optics and determining the inclination in reference to the position of the area of interest. The conducted research revealed that LEO with an inclination equal to the border attitude of the considered area of interest ensured larger coverage and probability of acquiring imagery than the heliosynchronous orbit that is typically used in imaging missions. The article discusses four different variants of positioning the satellites in constellations that met the fundamental requirements of the mission in terms of the revisit time and coverage of the area of interest. Finally, the variant consisting of four satellites placed on four orbits, including two reverse orbits, as proposed by the authors, was found to be the most efficient. The constellation achieved the required coverage for all rivers as early as in the first day (after 12 h 53 min), and the imaging access was evenly distributed during the day. The satellite images acquired by the proposed constellation may significantly support the monitoring of water quality in the southern catchment of the Baltic Sea by increasing the frequency of measurements and providing information on any section of the river within a short time.

There are many opportunities for further development in research on monitoring the quality of inland waters. One of the main areas of research is the development of remote sensing technologies that will enable the quick detection and localization of pollution sources. The presented research results focus on the design of satellite constellations that will complement traditional in situ point measurements. Such a satellite system could significantly increase awareness of the current state of surface water quality not only in the southern Baltic Sea catchment, but also in inland waters and other water bodies, minimizing the effects of pollution and reducing the costs associated with their disposal. Future research will focus on the development of optimal orbit models for the described satellite constellations for flood risk monitoring.

## Figures and Tables

**Figure 1 sensors-23-06192-f001:**
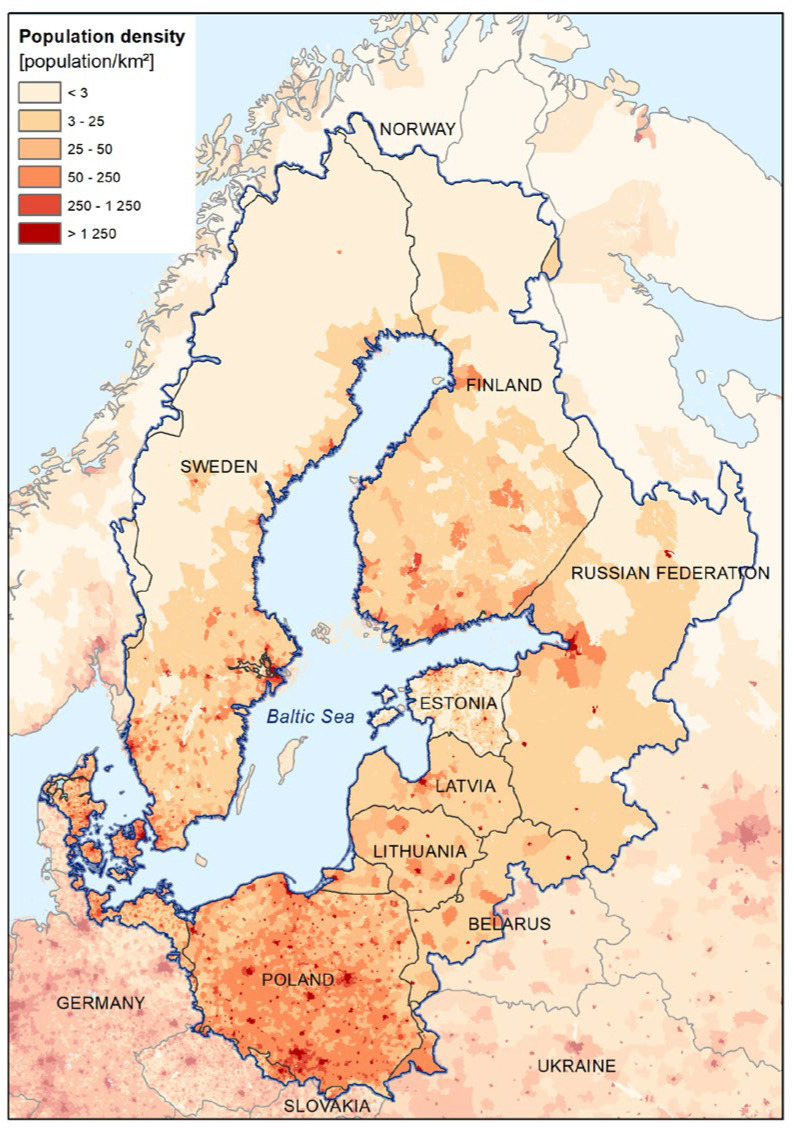
Catchment of the Baltic Sea and the population density in the area.

**Figure 2 sensors-23-06192-f002:**
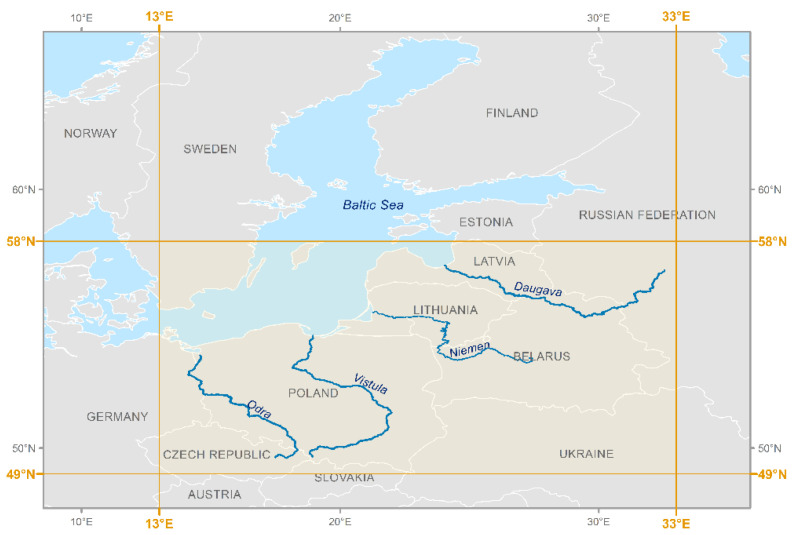
Longitudinal and latitudinal reach of the area of interest and the main rivers.

**Figure 3 sensors-23-06192-f003:**
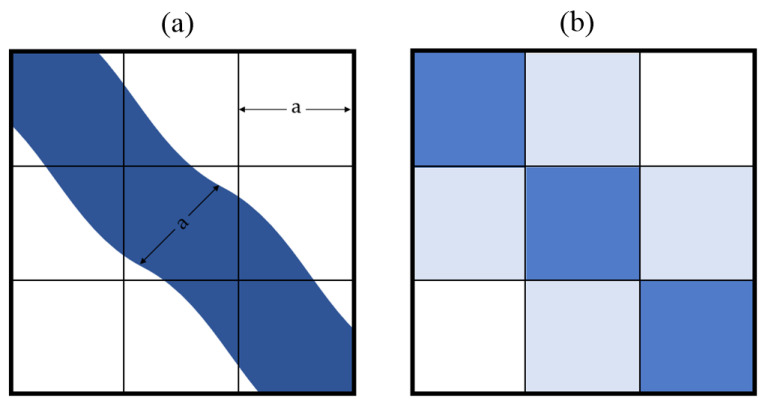
The problem of averaging the value of the pixel: (**a**) river whose width equals one times the GSD; (**b**) expected representation of the river in the image.

**Figure 4 sensors-23-06192-f004:**
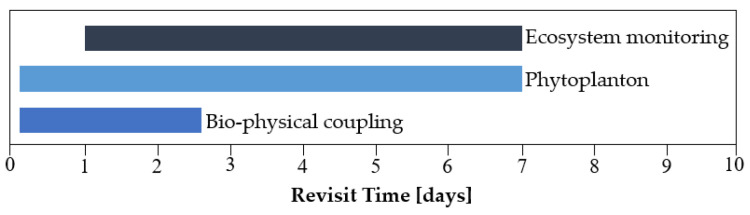
Period of revisit for remote sensing missions used in water management.

**Figure 5 sensors-23-06192-f005:**
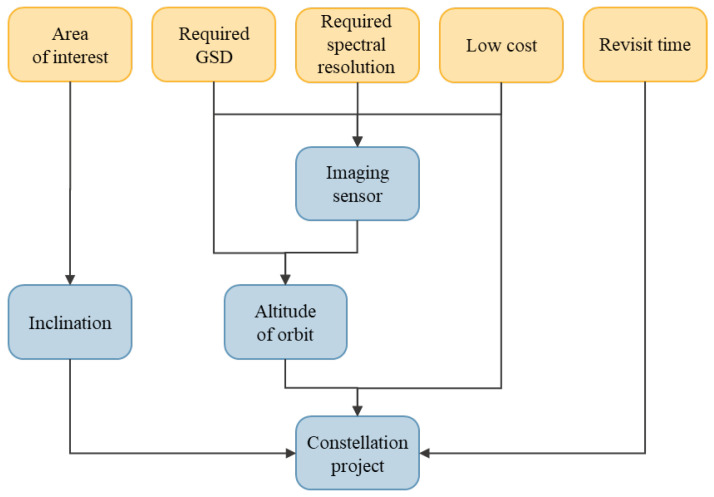
Flow chart of designing the constellation of satellites.

**Figure 6 sensors-23-06192-f006:**
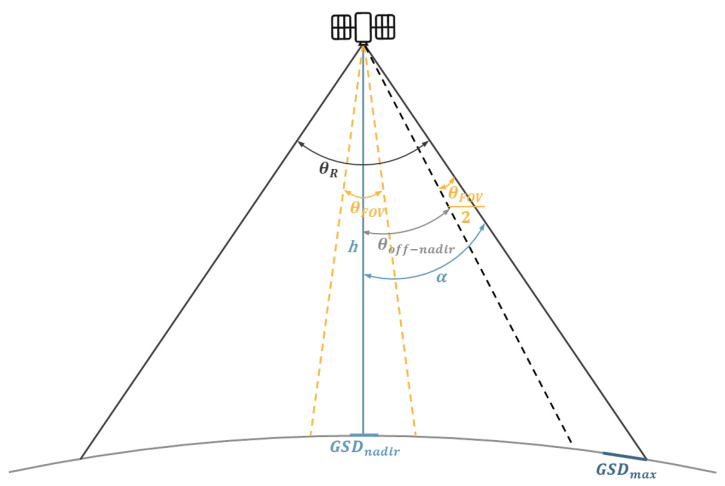
Values that are considered for determining the spatial resolution and orbit altitude.

**Figure 7 sensors-23-06192-f007:**
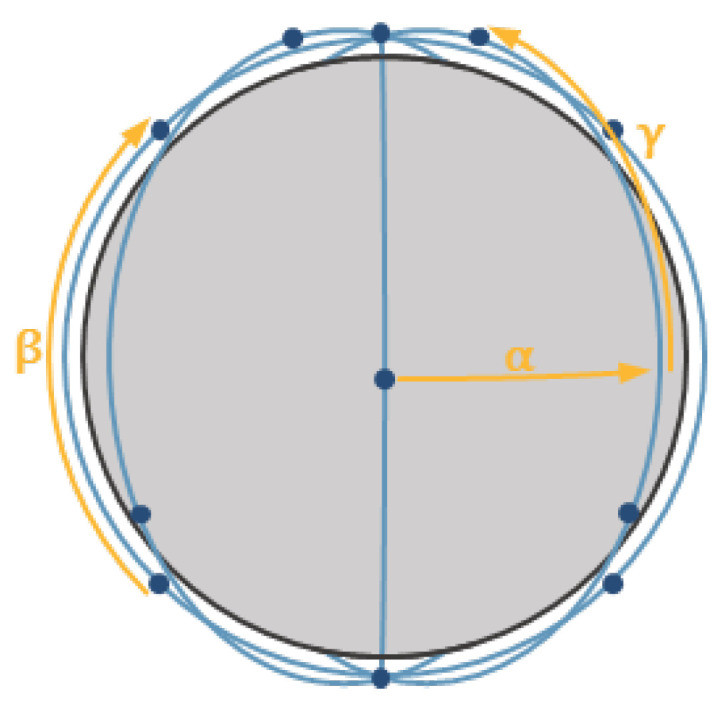
Angles α, β, γ in the constellation Walker Delta 90°: 16/4/1.

**Figure 8 sensors-23-06192-f008:**
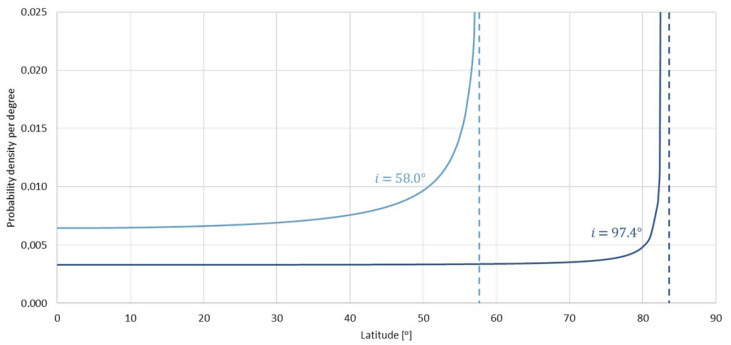
Density of probability distribution for the latitude for inclinations 58.0° and 97.4°.

**Figure 9 sensors-23-06192-f009:**
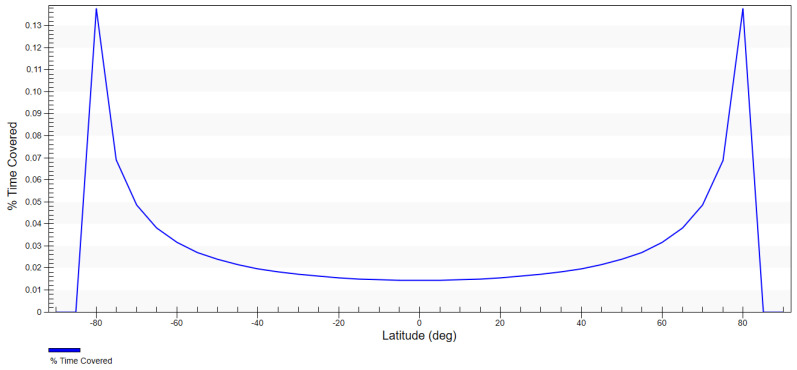
Percentage of time of coverage for the satellite of an inclination of 97.4° depending on latitude.

**Figure 10 sensors-23-06192-f010:**
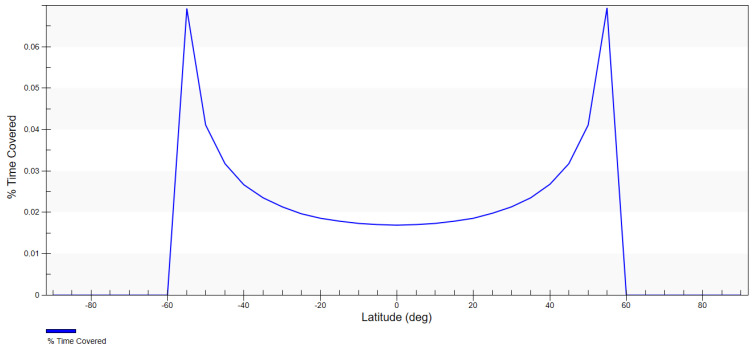
Percentage of time of coverage for the satellite at an inclination of 58.0° depending on latitude.

**Figure 11 sensors-23-06192-f011:**
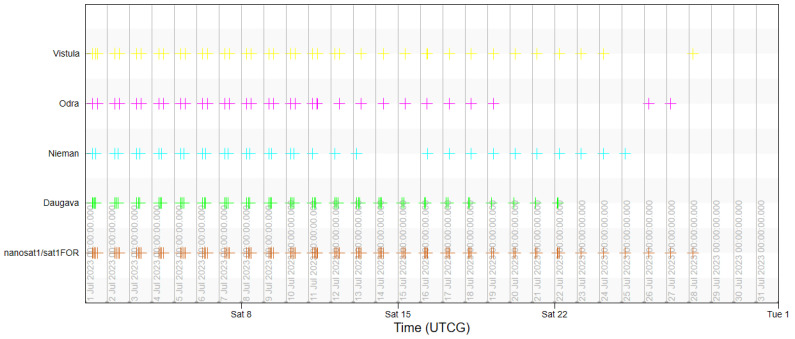
Access of a single satellite to imaging the main rivers with use of FOR.

**Figure 12 sensors-23-06192-f012:**
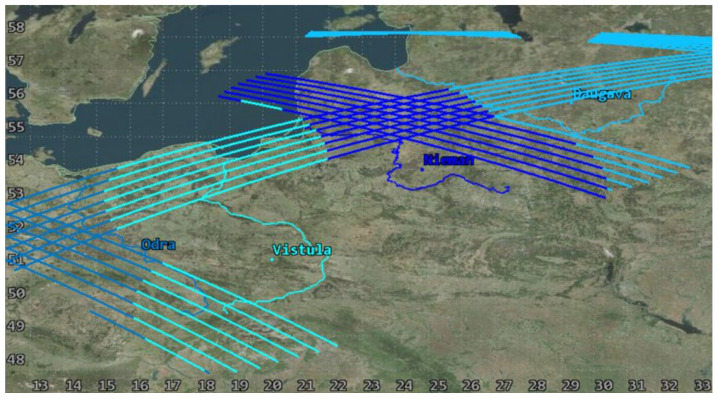
Ground tracks of a satellite during access to imaging the main rivers in the period from 1 July 2023 00:00:00 UTCG to 8 July 2023 00:00:00 UTCG.

**Figure 13 sensors-23-06192-f013:**
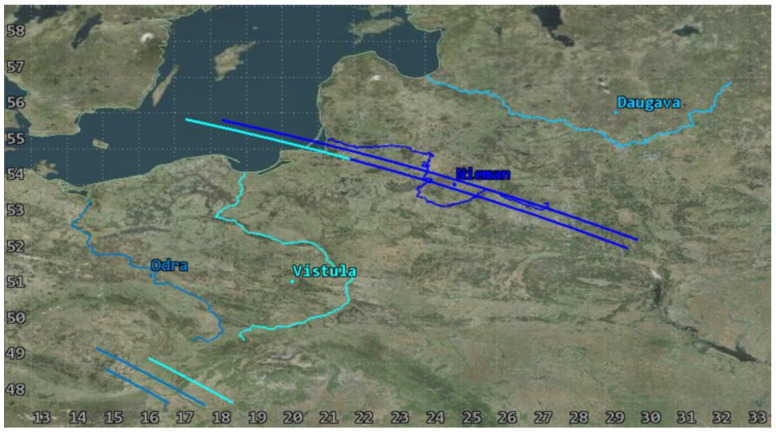
Ground tracks of a satellite during access to imaging the main rivers in the period from 24 July 2023 00:00:00 UTCG to 1 August 2023 00:00:00 UTCG.

**Figure 14 sensors-23-06192-f014:**
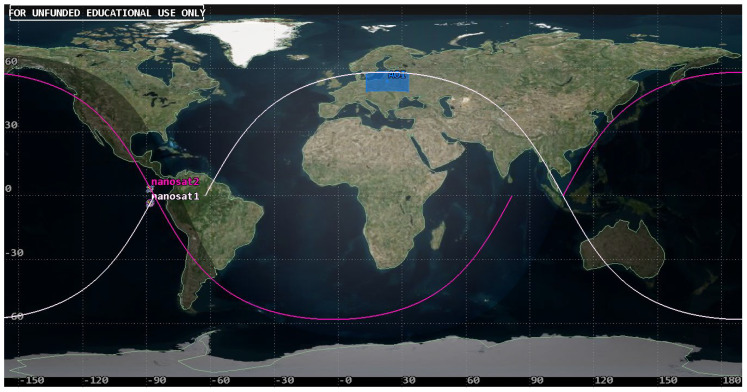
Presentation of the positioning of nanosatellites in Project 1 constellation and the area of interest (AOI).

**Figure 15 sensors-23-06192-f015:**
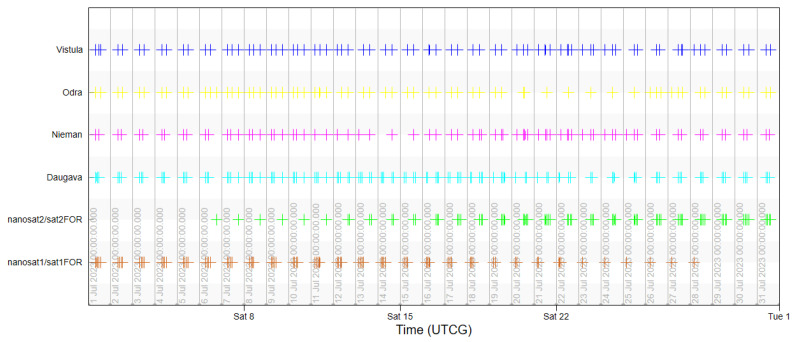
Access of two satellites in a constellation to imaging the main rivers with use of FOR.

**Figure 16 sensors-23-06192-f016:**
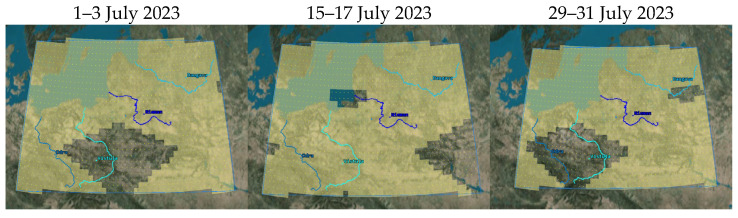
Accessibility of imaging of the area of interest for Project 1 constellation, determined with the use of FOR.

**Figure 17 sensors-23-06192-f017:**
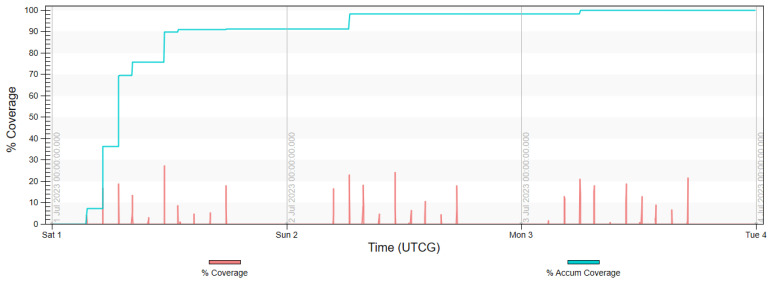
Imaging access and percentage of coverage of all rivers with use of FOR for Project 2.

**Figure 18 sensors-23-06192-f018:**
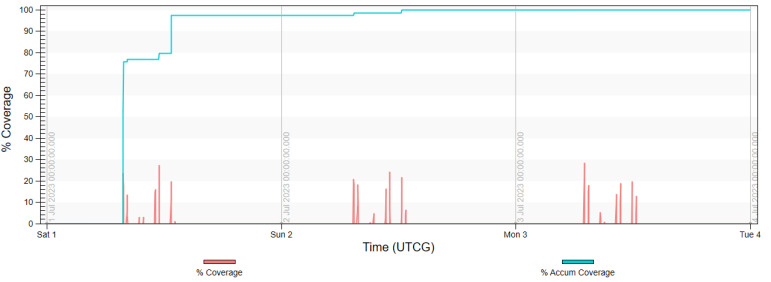
Imaging access and percentage of coverage of all rivers with use of FOR for Project 3.

**Figure 19 sensors-23-06192-f019:**
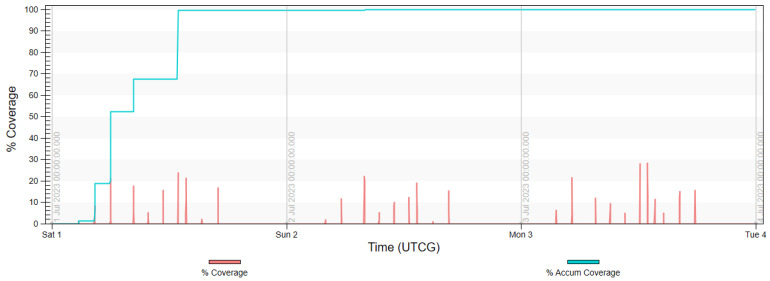
Imaging access and percentage of coverage of all rivers with use of FOR for Project 4.

**Figure 20 sensors-23-06192-f020:**
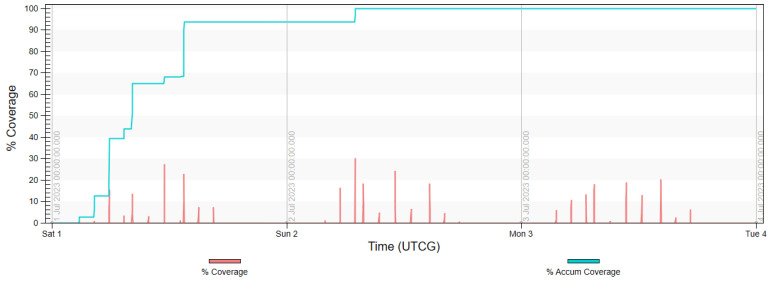
Imaging access and percentage of coverage of all rivers with use of FOR for Project 5.

**Figure 21 sensors-23-06192-f021:**
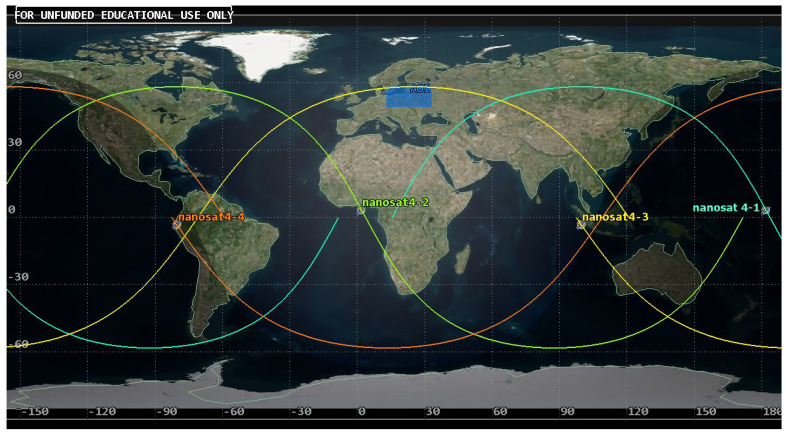
Presentation of the positioning of nanosatellites in Project 4 constellation and the area of interest (AOI).

**Table 1 sensors-23-06192-t001:** Range of width of the main rivers depending on the section.

River Name	River Bed Width		
	Upstream	Midstream	Downstream
Odra	10–70 m	50–250 m	100–200 m
Vistula	20–250 m	200–1500 m	250–1000 m
Niemen	10–100 m	80–250 m	200–300 m
Dźwina	20–100 m	70–300 m	120–280 m

**Table 2 sensors-23-06192-t002:** The objectives and requirements of the satellite mission.

Type of Orbit	LEO, Circular
Size of the area of interest	13° E–33° E, 49° N–58° N
Spatial resolution—(GSD)	5 m
Spectral resolution	minimum 5 channels in the VNIR range
Revisit period	3 days

**Table 3 sensors-23-06192-t003:** Main parameters of the considered sensors [32,33,34].

	Caiman Imager	MultiScape100 CIS	HyperScape100
GSD_nadir_ for 500 km	PAN: 3.0 m MS: 6.0 m	4.75 m	4.75 m
Swath_nadir_ for 500 km	12 km	19.4 km	19.4 km
Spectral resolution	7 VNIR channels	7 VNIR channels	32 VNIR channels
Radiometric resolution	8 or 16 bit	10 bit	10 bit
Focal length	-	580 mm ± 1 mm	580 mm ± 1 mm
FOV	-	2.22° × 1.66°	2.22° × 1.66°
Pixel size	-	5.5 μm	5.5 μm
Weight	1.8 kg	1.1 kg	1.1 kg
Dimensions	10 × 10 × 23 cm	9.8 × 9.8 × 17.6 cm	9.8 × 9.8 × 17.6 cm
Platform size	min. 6 U	min. 3 U	min. 3 U

**Table 4 sensors-23-06192-t004:** Data used in calculations and the calculated values.

Data	Calculated Values
${GSD}_{max}=5\;m$	α=21.11°
$\theta_{off-nadir}=20{}^{\circ}$	h=491.88 km
$\theta_{FOV}=2.22{}^{\circ}$	${GSD}_{nadir}=4.66\;m$
f=580 mm	${Swath}_{nadir}$ = 19.1 km
px=5.5 μm	${Swath}_{off-nadir}$ = 21.6 km

**Table 5 sensors-23-06192-t005:** Percentage of time of coverage of latitudes depending on inclination.

Latitude	Inclination	
	97.4°	58.0°
49°	0.02%	0.04%
55°	0.03%	0.07%
58°	0.03%	0.03%

**Table 6 sensors-23-06192-t006:** Accessibility of river imaging for Project 1 constellation, determined with the use of FOR.

River Name	Analysed Period		
	1–3 July 2023	15–17 July 2023	29–31 July 2023
Odra	100.00%	100.00%	21.94%
Vistula	44.86%	100.00%	9.03%
Niemen	100.00%	87.64%	100.00%
Daugava	100.00%	100.00%	74.08%

**Table 7 sensors-23-06192-t007:** Parameters of nanosatellites in constellations.

Satellite	Constellation											
	Project 2			Project 3			Project 4			Project 5		
	i	Ω	TA	i	Ω	TA	i	Ω	TA	i	Ω	TA
nano1	58°	0°	0°	58°	0°	0°	58°	90°	180°	58°	0°	0°
nano2	58°	90°	0°	58°	0°	90°	58°	270°	180°	58°	180°	90°
nano3	58°	180°	0°	58°	180°	180°	122°	180°	0°	122°	90°	180°
nano4	58°	270°	0°	58°	180°	270°	122°	0°	0°	122°	270°	270°

**Table 8 sensors-23-06192-t008:** Imaging accesses of the constellation to the rivers with use of FOR in the period 1–3 July 2023.

River Name	Project 2	Project 3	Project 4	Project 5
Odra	12	12	11	12
Vistula	18	14	15	14
Niemen	17	12	17	15
Daugava	30	24	31	34

**Table 9 sensors-23-06192-t009:** Time of achieving accumulated coverage of 95%.

River Name	Project 2	Project 3	Project 4	Project 5
Odra	6 h 48 min	12 h 42 min	12 h 53 min	7 h 0 min
Vistula	54 h 01 min	36 h 19 min	12 h 53 min	30 h 59 min
Niemen	11 h 29 min	11 h 29 min	6 h 0 min	5 h 50 min
Daugava	5 h 11 min	7 h 49 min	8 h 19 min	11 h 30 min

## Data Availability

Not applicable.

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
