# Peer review of "Design and Performance Analysis of a Constellation of Nanosatellites to Monitor Water Quality in the Southern Catchment of the Baltic Sea"

_sensors, 2023, doi:10.3390/s23136192_

Round 1

Reviewer 1 Report

I am happy to review the manuscript. Authors are clearly addressed the importance of nanosatellite sensor to monitor inland water quality in the introduction section with comparing others sensors images. They have testes the symmetric constellation projects in different time period and found a project suitable for only Baltic sea. I have some suggestions or queries as bellows:

1.       Is this symmetric constellation applicable only for the Baltic sea? Or it can be used for other inland waters too.

2.       I did not find here the applicability of the project to retrieve any water quality parameters of Baltic sea.

3.       Discussion part is missing here. Authors should discuss some of their finding with other sensors work such as Sentinel Image (10 m spatial resolution) in case of water quality parameter retrieval.

4.       Lastly, the data availability of the work need to address.

Overall, the manuscript seems quite interesting. I would like to accept the manuscript after addressing the above mentioned comments.

Author Response

We are very grateful for your valuable comments, which have helped us improve this article and provided useful suggestions for the future.

Answering the questions:

  1. The satellite constellation project is dedicated to monitoring the quality of inland waters in the southern catchment area of the Baltic Sea, especially the main rivers. It can be used to assess the quality of surface waters at latitudes not greater than 58° (due to the applied orbital inclination). Still, the effectiveness, including the achieved revisit time for other regions, has not been tested. The proposed methodology for designing a satellite mission is universal and can be used to develop a constellation dedicated to other inland waters.
  2. The satellite constellation was designed to provide remote sensing data in the form of images, based on which it is possible to estimate water quality parameters such as transparency, turbidity, chlorophyll content, etc. That issue was analyzed based on literature examples in Chapter 2, which was then the basis for imaging sensor selection. The work did not address the subject of further data processing to obtain water quality parameters.
  3. The work was aimed at the selection of orbit parameters for satellite constellations. Among the analyzed ones, the constellation concept consisting of four satellites placed in four orbits (including two retrograde ones) was selected. It is strictly dedicated to observing the southern catchment of the Baltic Sea. For this reason, it was not compared with existing commercial solutions, which are designed for many applications.
  4. The suitability of the proposed constellations was tested by simulation. Expected data availability was analyzed through imaging capabilities considering constraints such as lighting conditions.

with best regards

Piotr Walczykowski

Reviewer 2 Report

This is an interdisciplinary paper related with agriculture, satellite imagery, sustainability, remote sensing. Authors focus on an important aspect related with the lives of every human person on the planet. They present results of studies including a number of small sats, this topic is surely worth investigating, particularly in the Baltic Sea region.

To start with, the manuscript is written in proper English, it is informative and pleasant to read. The introduction provides a good background, followed by appropriate citations.

There are some minor editorial and formatting issues, e.g., lack of space or multiple space signs between subsequent words, typos, etc. Additionally, check how to properly use Capital Letters in case of Chapter Titles. Next, check the size, layout and enumeration of respective symbols and mathematical equations and units (like degrees). Check the indentation of some paragraphs and figure/table captions, as well as enumeration of Chapters (e.g., see Conclusions). Therefore, a careful throughout examination would seem necessary.

Remember to always refer to a Figure in text before discussing its contents. The case of, e.g., Fig. 1 and 2, is such an example.

Fig. 4 is simply too large, the fonts get blurred.

Check the enumeration of figures and tables, and references to them in text, i.e., what is and where is Figure 4.2, 4.4, 4.5, 4.6, etc.? where is Table 4.4, 4.5, etc.?

Could the Authors provide at least principle operation considering the utilized frequency band, channel width, bitrate, throughput, latency, modulation scheme, when it comes to the communication link?

The Conclusions section could be extended. Do provide additional comments and feedback on your findings, do mention about open issues and future study directions.

Taking into consideration the area of interest, it would seem advisable to extend the number and scope of cited references.

To sum up, this is a good paper, yet it deserves to be a very good one. Authors are advised to prepare a revised version of their manuscript.

There are some minor editorial and formatting issues, e.g., lack of space or multiple space signs between subsequent words, typos, etc. Additionally, check how to properly use Capital Letters in case of Chapter Titles. Next, check the size, layout and enumeration of respective symbols and mathematical equations and units (like degrees).

Author Response

Dear Reviewer,

Thank you for your valuable comments, which have helped us to improve this article and provided valuable suggestions for the future.

Editorial and formatting errors, including numbering and references to figures, tables, size of figures, formula formatting, etc., have been corrected as suggested. The content has been extended with communication issues. Literature has been added.

With best regards

Piotr Walczykowski

Round 2

Reviewer 1 Report

I am happy with the response of authors. I think it is now in right form to accept the manuscript.

Author Response

Dear Reviewer,

Once again, thank you very much for your comments.

Sincerely

Piotr Walczykowski

Reviewer 2 Report

Authors have prepared a revised version of their paper, which has increased the overall quality of the initial submission. Now, the text is more informative, yet changes made are very brief and cursory.

In the future, do remember about uploading a response to the Reviewer in a separate file, along with justification on doing or not doing something regarding the proposed modifications, comments, etc.

Still, there are some minor editorial and formatting issues, including typos, particularly when it comes to referring to figures or tables in the text itself, as well as figure/table captions, not to mention the use of Capital letters. The ‘style’ of a typical paper published in Sensors is available in Open Access, you can access it anytime and examine how it is prepared. Remember to always make a careful throughout examination before resending your paper to the system.

Also, do check the e-mail addresses of all co-authors, i.e., there is a typo in [student] in case of one of them.

At the end, check once again the proper way to format and prepare the list of cited references.

This work has potential, it is a good paper, yet it deserves to be a very good one. Taking all the above into consideration, this manuscript requires a second minor round of revisions.

Author Response

Dear Reviewer,

Thank you very much for your valuable comments. Editorial and formatting problems, references to figures and tables were corrected in the final text. A typo in the email address has also been corrected. The cited literature has been reformatted again.

Once again, thank you very much for your comments.

Sincerely

Piotr Walczykowski